# DInSAR for Road Infrastructure Monitoring: Case Study Highway Network of Rome Metropolitan (Italy)

**Felipe Orellana** [1,*], **Jose Manuel Delgado Blasco** [2], **Michael Foumelis** [3], **Peppe J.V. D'Aranno** [4], **Maria A. Marsella** [1] and **Paola Di Mascio** [1]

1   Department of Civil, Building and Environmental Engineering, Via Eudossiana 18, Sapienza-University of Rome, 00186 Rome, Italy; maria.marsella@uniroma1.it (M.A.M.); paola.dimascio@uniroma1.it (P.D.M.)
2   Grupo de Investigación Microgeodesia Jaén (PAIDI RNM-282), Universidad de Jaén, 23071 Jaén, Spain; j.dblasco@ujaen.es
3   BRGM—French Geological Survey, 45060 Orleans, France; m.foumelis@brgm.fr
4   Survey Lab, Spin-off–Sapienza, Via Eudossiana 18, 00184 Roma, Italy; peppe.daranno@surveylab.info
*   Correspondence: felipe.orellana@uniroma1.it

**Abstract:** The road network of metropolitan Rome is determined by a large number of structures located in different geological environments. To maintain security and service conditions, satellite-based monitoring can play a key role, since it can cover large areas by accurately detecting ground displacements due to anthropic activities (underground excavations, interference with other infrastructures, etc.) or natural hazards, mainly connected to the critical hydrogeological events. To investigate the area, two different Differential Interferometry Synthetic Aperture Radar (DInSAR) processing methods were used in this study: the first with open source using the Persistent Scatterers Interferometry (PSI) of SNAP-StaMPS workflow for Sentinel-1 (SNT1) and the second with the SBAS technique for Cosmo-SkyMed (CSK). The results obtained can corroborate the displacement trends due to the characteristics of the soil and the geological environments. With Sentinel-1 data, we were able to obtain the general deformation overview of the overall highways network, followed by a selection and classification of the PSI content for each section. With Cosmo-SkyMed data, we were able to increase the precision in the analysis for one sample infrastructure for which high-resolution data from CSK were available. Both datasets were demonstrated to be valuable for collecting data useful to understand the safety condition of the infrastructure and to support the maintenance actions.

**Keywords:** DInSAR; infrastructure monitoring; road subsidence; Rome

## 1. Introduction

The quantitative assessment of ground displacement affecting infrastructure is traditionally based on ground instrumentation, using automatic total stations, ground, and mobile laser scanners, with sub-centimeter precision that are useful for maintenance processes [1]. Space-borne Differential SAR Interferometry (DInSAR) is an alternative solution that can be fully assimilated into ground monitoring. Civil infrastructure monitoring with DInSAR is relatively new and is currently not being fully exploited. The large amount of data available is acquired more frequently and with high precision at low costs; in the case of Sentinel-1, these aspects make it an attractive source of information [2]. The technology used in this work is based on advanced DInSAR approaches [3–6], which consists of exploiting SAR acquisition sequences collected over long periods of time, acquired in the same geometry, and allows us to extract useful information on the spatial and temporal patterns of displacement detected through the generation of time series, with precision of a centimeter to

millimeter [7,8]. The DInSAR technique is an extension of the InSAR technique and allows measurement of sub-centimeter ground displacement, with millimeter precision, using the phase difference between series of SAR images that it acquires at different times on the same scene [9].

DInSAR displacement time series have been largely exploited in a wide variety of geophysical contexts, such as seismic, volcanic, and mass movement scenarios, with a dual objective: to map and monitor detected displacements for large areas [10–14]. One of the main attractions of satellite-based DInSAR is its ability to cover very large areas at a systematic and continuous pace remotely, making it suitable as a monitoring and control tool in a transport infrastructure network.

Structural and infrastructural health monitoring with the DInSAR technique is becoming one of the most powerful and economic means [15,16]. DInSAR observations from satellites are becoming increasingly reliable for long-term and wide-area displacement monitoring. The main contribution of this technique is the exploitation of a large number of observed point displacements, distributed throughout the entire structure [17]. The first applications in the field of infrastructure date back more than two decades. Currently, DInSAR is used to study curettage phenomena and structural failures through historical SAR analysis. The pre-collapse spatial geodetic observations of the Morandi bridge, in [18], propose a methodology for the evaluation of the collapse, based on observations of Synthetic Aperture Radar (SAR) and the Markov-Chain-Monte-Carlo (MCMC) approach, was applied to the bridge, generating an integral-multi-sensor displacement map of time series, from measurements based on a historical analysis of SAR images, acquired by the Cosmo-SkyMed constellation and the Sentinel-1A/B constellation. Although many studies have been carried out using DInSAR in the field of monitoring individual infrastructures, such as bridges [18–21], railways [22,23], and subways [24–26], few documents use DInSAR for infrastructure network monitoring and maintaining [27,28].

In this study, the movements of line-of-sight (LOS) of the urban highways in Rome (namely A24, A12, A90, A91, and the A1 with its branches D18-D19) were analyzed. The sections of the highways with the highest subsidence rates were selected and classified on the basis of PSI results. Each section was divided in samples 100 m long, and for each sample, the average velocity (mm/year) with the internal PSI was calculated. In addition, by means of the hydrogeological map of Rome (La Vigna et al., 2015), it was possible to relate the subsidence to the soil type and use some rainfall data in certain areas. The intention of applying Sentinel-1 data was to obtain an overview of the displacements observed along the infrastructure network and to analyze the performance of Sentinel-1 data in three areas showing major subsidence, one of which was also investigated by using Cosmo-SkyMed to deepen the analysis.

Two processing techniques were used: StaMPS integrated with SNAP [29] and the SBAS technique [30]. Two types of SAR images were also used, the first obtained from the Sentinel-1 sensor of ESA (European Space Agency) and the second images obtained from Cosmo-SkyMed of ASI (Agency Italian Space).

*1.1. Study Area*

The study area is located at central Italy (Lazio region), specifically at the metropolitan area of Rome, and it includes the area covered by the road network of the metropolitan area. To assess the structural condition of the road network, we initially covered the entire network and then we deepened the analysis on the highway with the highest subsidence. Using the Sentinel-1 data, an overall overview of the deformation pattern on the extension of the highway network of the metropolitan area of Rome, which includes the northern branch D18, the A1 highway, connecting Rome with the A90 "Grande Raccordo Anulare" (GRA), and the A24, was developed. In addition, by using Cosmo-SkyMed data, the analysis was deepened to the infrastructure the A91 "Rome–Fiumicino" running on an embankment and composed of two separated carriageways with two lanes each. A91 is one of the busiest roads in Rome, since it connects the city to the international airport.

## 1.2. Geological Setting

The region's geology is characterized by volcanic deposits (mainly pyroclastic tuff) from the Albano volcano district to the southeast and the Sabatini volcano district to the northwest, with alluvial sediments along the Tiber valley [31]. The topography gradually decreases from these two volcanic districts towards the Tiber, with valleys carved by the erosion of the river. The region was founded in the Paleolithic period during the period of ancient Rome and, more recently, during the recovery of the Ostia and Maccarese ponds between the end of the 19th century and the beginning of the 20th century. The stratigraphic and paleoenvironmental evolution of the Tiber Delta occurs during the late Pleistocene and Holocene; this has been reconstructed by numerous studies that integrate stratigraphic, micropaleontological, geomorphological, and archaeological studies dating from the 14th century [32–35].

As already mentioned, the investigated region is located in the city of Rome and its surroundings; this city has a historical connection with water, and its large aqueducts, fountains, rivers, and the sea have defined the characteristics of its environment. The Tiber delta has been interpreted as being dominated by waves [36–38]. The hydrogeological information reference base for this work is the hydrogeological map of the city of Rome, scale 1: 50.000 [39], shown in Figure 1. Hydrogeological formations have been defined in hydrogeological complexes, each characterized for the same transmission and storage capacities, as well as for a similar hydrogeological significance with respect to groundwater circulation at the scale of the entire study area.

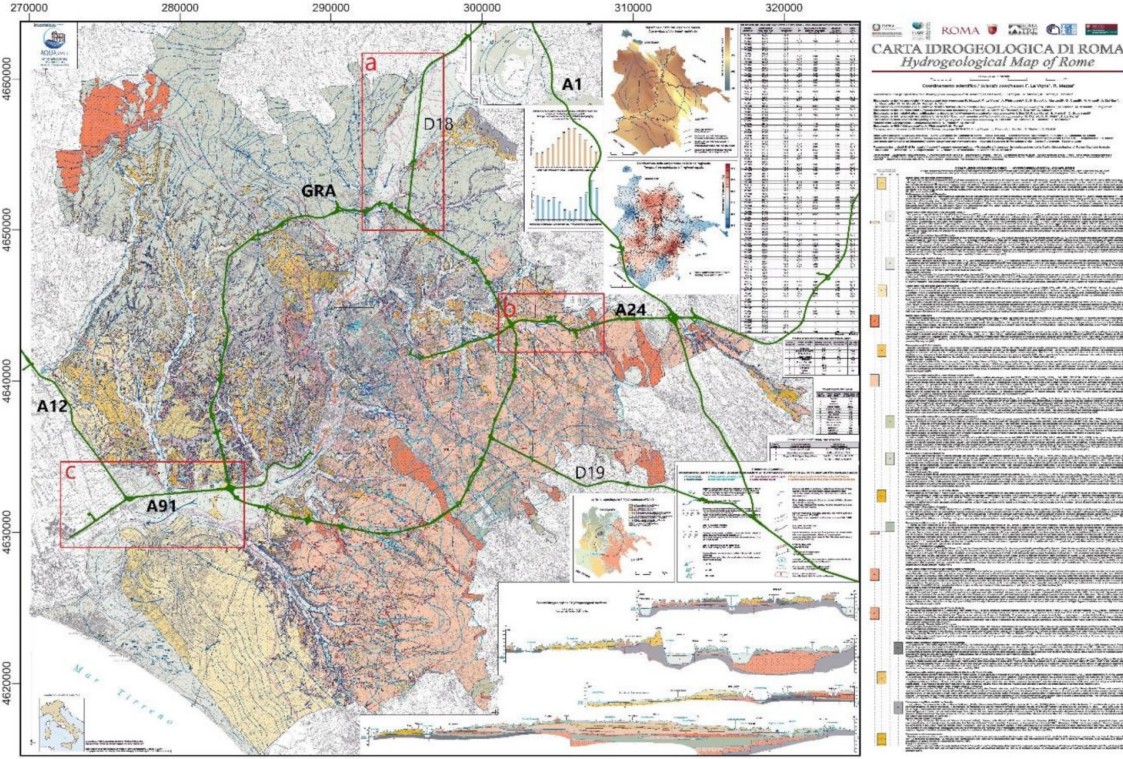

**Figure 1.** Highway network route of Metropolitan Rome, as a background Hydrogeological map of Rome (La Vigna et al., 2015).

The hydrographic network of Rome consists of two main elements, the river that runs through the city and the entire metropolitan territory, encompassing the highways network. To analyze the subsidence of the road network, it is possible to relate three areas (a), (b), and (c) shown in Figure 1. In area (a), we can mainly find alluvial deposits of the Tiber river; in area (b), volcanic and alluvial deposits of the Aniene river; and in area (c), alluvial deposits of the left of the Tiber delta.

The hydrogeological environment of the Tiber Delta has a deep and artesian main aquifer located in sand gravel, and it is supported at the base by clays from the lower Pleistocene that act as an aquifer and at the top partially sealed by deposits of clay and limestone of low permeability [40]. Another aquifer is found in sandy deposits; its piezometry has been reconstructed to the left of the Tiber River [41,42].

## 2. Materials and Methods

### 2.1. Dataset

The Copernicus Sentinel-1 SAR data used covered the period 30 March 2015–13 April 2018 using descending geometries. We limited our analysis to Sentinel-1A data only (12-day repeat cycle) (Table 1), which is considered sufficient given the expected magnitude of ground displacements and the availability of a large archive over the area of interest. It should be noted that since Sentinel-1 products are not spatially synchronized, meaning that their start and end times can vary within each orbit, more than one scene is often required to fully cover our area of interest [43]. The advantages of Sentinel-1 come from its wide range coverage (250-km swath in the interferometric wide mode) and sufficient spatial resolution (5 m × 20 m in range vs. azimuth). In addition, the TOPS (Terrain Observation with Progressive Scans) imaging technique guarantees homogeneous image quality throughout the range [44].

The high-resolution SAR data used are from the Cosmo-SkyMed mission, and covered the period 29 July 2011–7 March 2017. The Constellation of the Italian Space Agency (ASI) allows a 1-6-11 day repeat cycles for the same area of Earth with the same acquisition mode with each satellite. The products used were Stripmap-HIMAGE (3 m × 3 m in range vs. azimuth). One of the advantages of the X-band is the high spatial resolution, and also the nominal (full size) constellation orbit configuration was designed to achieve the best balance between cost and performance, providing global access to Earth at a constellation level of a few hours, with at least two opportunities in one day to access the same target site on Earth under different observation conditions [45].

The area of interest (AOI) and the extent of the ascending and descending Sentinel-1 orbits and the descending Cosmo-SkyMed orbit are illustrated in Figure 2 and Table 1 indicates the observation periods for each sensor.

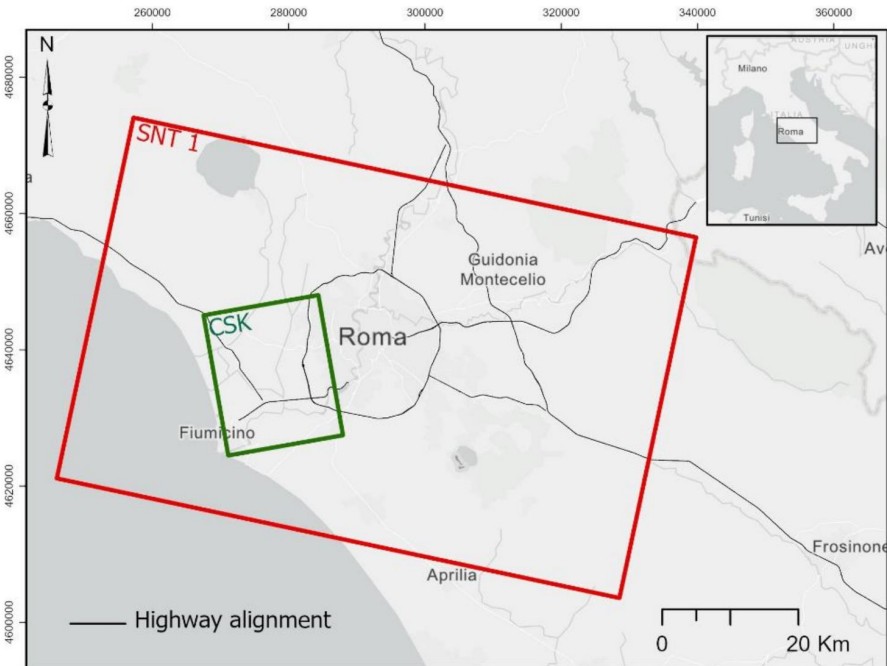

**Figure 2.** Area of interest and footprint of the selected Sentinel-1 tracks and Cosmo-SkyMed tracks.

**Table 1.** Sentinel-1 data employed for study, with first and last image of each dataset, band, orbit, and number of acquisitions.

| Sensor | First Image | Last Image | Band | Orbit | N° Acquisitions |
|---|---|---|---|---|---|
| Sentinel-1 | 30 March 2015 | 13 April 2018 | C | 22 DSC | 82 |
| Cosmo-SkyMed | 29 July 2011 | 07 March 2017 | X | DSC | 68 |

*2.2. Interferometric Processing*

In this work, two interferometry techniques were used, the first based on a completely open source SNAP-StaMPS workflow [46], with open SAR data on the Copernicus Open Access Hub of the European Space Agency (ESA) [47]. The second was based on a Small Baseline Subset (SBAS) technique [8,30] using a Cosmo-SkyMed dataset obtained by a License Agreement with the Italian Space Agency (ASI).

2.2.1. SNAP-StaMPS PSI

The Sentinel-1 PSI dataset was processed in [43], and it is openly available [48]. Interferometric processing was carried out using the Sentinel Application Platform (SNAP) [49], developed by ESA under a GPL license v3. We employed the snap2stamps package, which [46] provides a series of scripts to automate the processing of entire stacks of SLC Sentinel-1 images in batch mode. The Stanford Method for PS (StaMPS), is a MATLAB-based software developed at Stanford University and subsequently updated at the University of Iceland, Delft University of Technology, and the University of Leeds. The package is equipped with persistent scatterer interferometry and small baseline methods and with an option to combine them. It is developed to make PSI processing work well on arid terrain in non-urban areas that undergo non-linear deformation without artificial structures. It uses amplitude and phase information from individual pixels to determine the probability of being a persistent scatterer (PS).

To automate the interferogram formation process, the "snap2stamps" package was used, which, depending on the configuration of some parameters (Sentinel-1 images sub-figure to be processed, bounding box of the area of interest, path to the data folder, the path and name of the master image, the parameters related to the computational resources to be used, etc.), the interferograms are calculated automatically. These scripts are Python wrappers that use SNAP as an InSAR processor and provide StaMPS compliant results for PSI analysis. Those sets of scripts are available as open source on GitHub [46].

Multi-temporal PSI processing was based on the StaMPS interferometric processing scheme [50]. The standard PInSAR technique [4] selects PS candidates based on their phase variation over time, assuming a temporarily linear strain model, while the StaMPS methodology exploits the spatial correlation of its phase without previous assumptions about its temporal nature [51]. Therefore, StaMPS is successful in identifying less bright dispersers with lower SCR (signal to clutter) and therefore provides denser PS coverage even on sparsely developed and natural terrain.

2.2.2. SBAS Technique

The SBAS technique implements an easy combination of SAR interferograms generated from an appropriate selection of SAR data pairs characterized by a small spatial and temporal baseline. The key objective of data selection is to mitigate the de-correlation phenomena, maximizing the number of pixels exploited.

Unlike other advanced multitemporal DInSAR techniques, the SBAS approach does not require any a priori information, representing a valuable method for detecting and measuring non-linear LOS displacement over time [52], providing displacement maps and time series.

The SBAS approach can analyze interferograms generated by a complex averaging operation, called complex multilooking, applied to full-resolution data. This is a relevant topic because the possibility of analyzing multilook interferograms allows a reduction of the amount of data to be

processed. Furthermore, distribute scatters (DSs) are created, thus increasing the coherence and simplifying the investigation of extended areas. It is worth noting that the multitemporal SBAS technique has been extensively evaluated in various deformation contexts, recovering an accuracy of approximately 5 mm for single deformation measurements and 1 mm/year for information on mean deformation velocity [7,53,54]. Such an approach was applied to process the CSK dataset collected from 2011 to 2017 over the A91 highway, providing the results described in Section 4.

## 3. Results 1: Sentinel-1 Data

### 3.1. Low Resolution Analysis: Highway Network of Rome Metropolitan

The general deformation overview is represented in Figure 3a, using the data from the PSI Sentinel-1. We proceeded to extract the target of the PSI into the highway alignment considering a corridor through a buffer, along the highway (see Figure 3b). Table 2 shows the amount of PSI contained by each highway where we used the alignment obtained in the WEBSIT portal of metropolitan Rome [55].

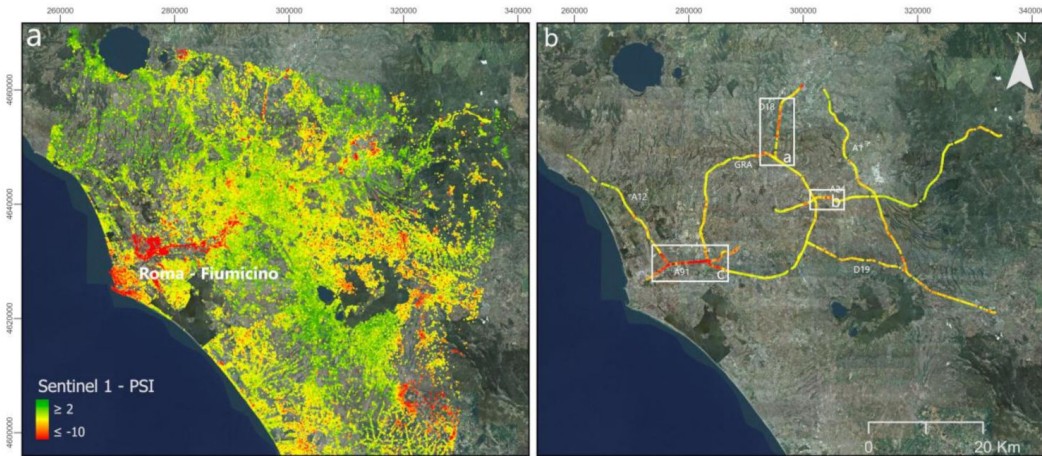

**Figure 3.** (**a**) Displacement map PSI Sentinel-1 for Rome metropolitan (**b**) Displacement map PSI for highway network, as background orthophoto Rome. The white insets highlight the 3-area selected for further analysis.

**Table 2.** Number of PSI target contained by each highway.

| Highway | Extension Detected (km) | N° PSI |
|---------|-------------------------|--------|
| A1 | 87 | 1641 |
| A12 | 35 | 313 |
| GRA | 68.5 | 3090 |
| A24 | 47 | 1161 |
| A91 | 18.5 | 461 |

In this analysis, three areas along the highways interested by subsidence were selected, as highlighted in Figure 3b: (i) the A91 (Roma-Fiumicino); (ii) the intersection D18 (A1 branch) with GRA (Grande Raccordo Anulare); and (iii) the A24 (Rome-Aquila). The maximum displacements are detected in the area on the Roma-Fiumicino highway where the PSI reaches more than a mean velocity of −15 mm/year.

### 3.1.1. Area (a): D18–GRA Highways

A deformation pattern was identified from the data obtained from Sentinel-1 and an area of intersection of two highways GRA and D18 was analyzed. The subsidence zones for this case are

indicated in boxes 1 and 2 of Figure 4a. We analyzed bridges and embankments located along the highways, superimposing the routes on the hydrogeological map of Rome [39], and it was found that the greatest displacement occurs in alluvial deposits and mainly in embankments near water courses (shown in Figure 4b).

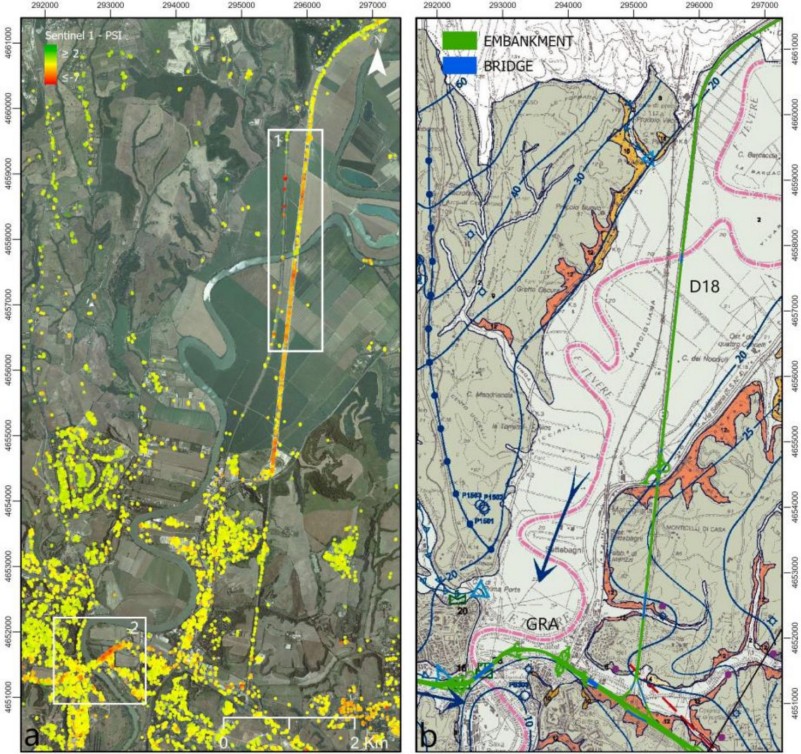

**Figure 4.** (**a**) Displacement map PSI Sentinel-1 and study area indicated in white boxes 1 and 2, as background orthophoto Rome, (**b**) Detail of the position of the D18 and GRA highway on the hydrogeological map of Rome (Vigna et al., 2015) shows the position of the route in the alluvial deposit in the northern area, southern zone volcanic deposit, (blue arrow) indicates the groundwater flow, (pink line) indicates the current course of the Tiber river.

For the section of the D18 highway (Box 1, Figure 4a), ~180 PSI were obtained with a standard deviation of 1.22, and PSI reaching −7.00 mm/year of displacement, and we proceeded to section every 100 mt and classify the PSI contained in each section of the road (see Figure 5a), obtaining the velocity mean of each section; in this case, we obtained a maximum velocity of −6.36 mm/year.

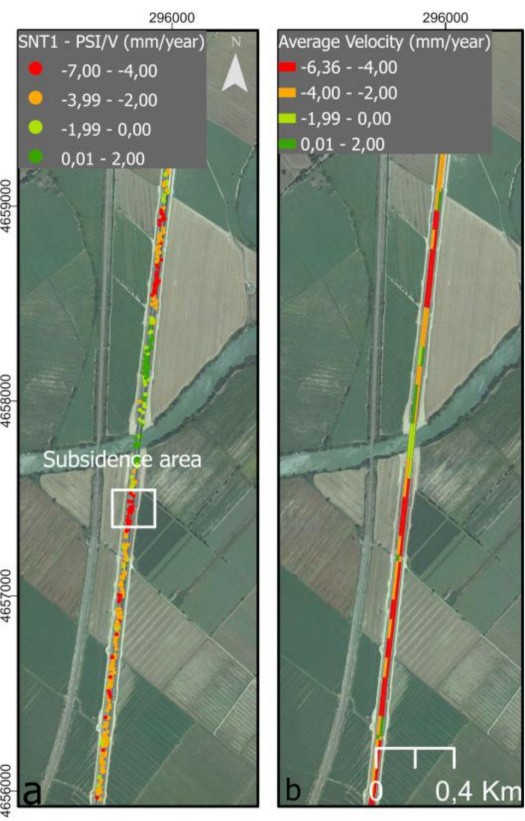

**Figure 5.** (**a**) Displacement map PSI Sentinel-1 for D18 highway, (**b**) Map classification section average velocity (mm/year), as background orthophoto Rome.

The time series of a representative section of the D18 highway is shown in Figure 5a. The tendency over a period of more than three years resulted in a cumulative displacement exceeding-20 mm (see Figure 6).

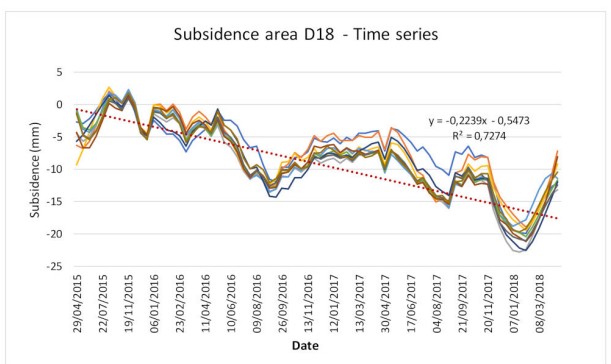

**Figure 6.** Subsidence area time series of the D18 highway.

For the section of the GRA highway (box 2, Figure 4a), we obtained ~150 PSI (Figure 5a) with a maximum velocity of −8.04 mm/year. In Figure 7b, the GRA sections were classified every 100 m, and the entry and exit routes were considered, since there was enough PSI. We calculated the average velocity (mm/year) of the PSI contained in each section dividing each lane, where we observed an average velocity of displacement of −6.32 mm/year close to the water courses.

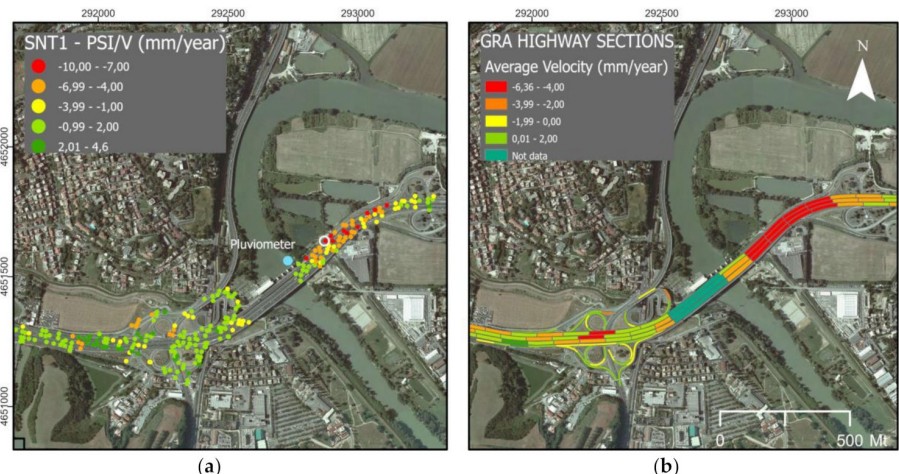

**Figure 7.** (**a**) Displacement map PSI Sentinel-1 for GRA highway (**b**) GRA sections, as background orthophoto Rome.

With the PSI displacement map (Figure 7a), we were able to identify the subsidence areas of the GRA, close to the water courses; therefore, the variation in soil stability can be caused by variations in the soil water and increased rainfall. With the precipitation data from the Castel Giubileo pluviometer, we were able to compare the precipitation data with the subsidence. The data of the rainy season used were extracted from the hydrological records of the Lazio region [56], during the satellite observation period. We were able to compare the subsidence with the rainfall, selecting the PSI (white point Figure 7a) with the highest subsidence with 8.04 mm/year and a coherence ≥ 0.90, and then we constructed the time series graph, as shown in Figure 8.

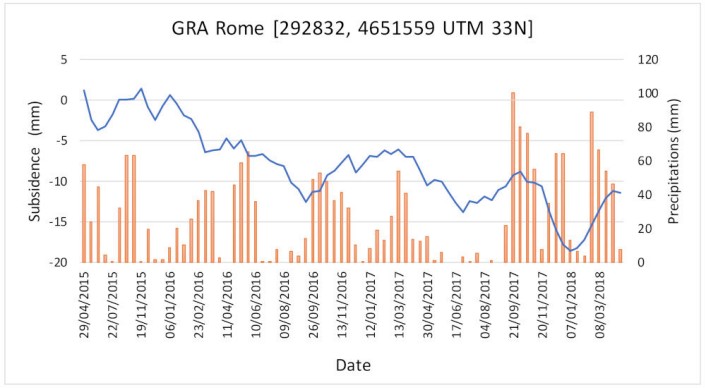

**Figure 8.** Graphic PSI target great subsidence, coherence ≥ 0.90 and precipitation cumulated "Castel Giubileo" pluviometer.

### 3.1.2. Area (b): A24 Highway

For the A24 highway (Rome–L'Aquila), we selected a nearby extra-urban area with geological complexity (see Figure 9b). In this area, a series of bridges and embankment sections are located and a considerable amount of data is available, where we could extract the PSI contained along the highway, to verify the good stability of the bridges and confirm subsidence in the embankments.

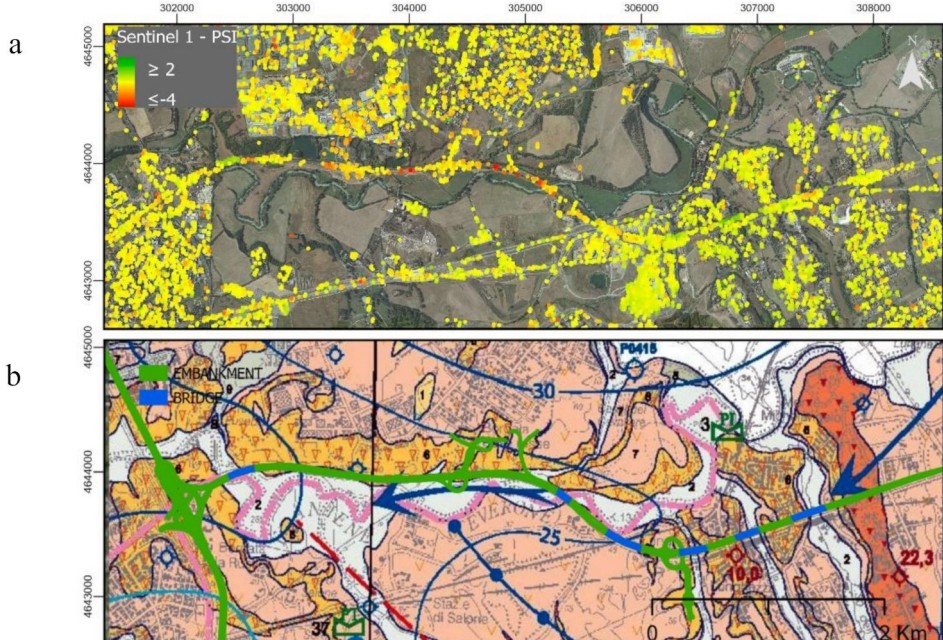

**Figure 9.** (**a**) Displacement map PSI Sentinel-1 for A24 Highway (Rome-Aquila), as background orthophoto Rome 2011. (**b**) The detail of the position of the A24 highway on the hydrogeological map of Rome (Vigna et al., 2015) shows the morphology, the position of the anthropogenic, and the volcanic deposit indicates that the flow of groundwater (blue arrow), indicates the current course of the Aniene (pink line).

The A24 motorway has undergone modifications in recent years; therefore, we used the ESRI source as a mapping base (Figure 10), and the construction of two parallel lanes, together with a series of reinforced concrete bridges, were the main modifications works.

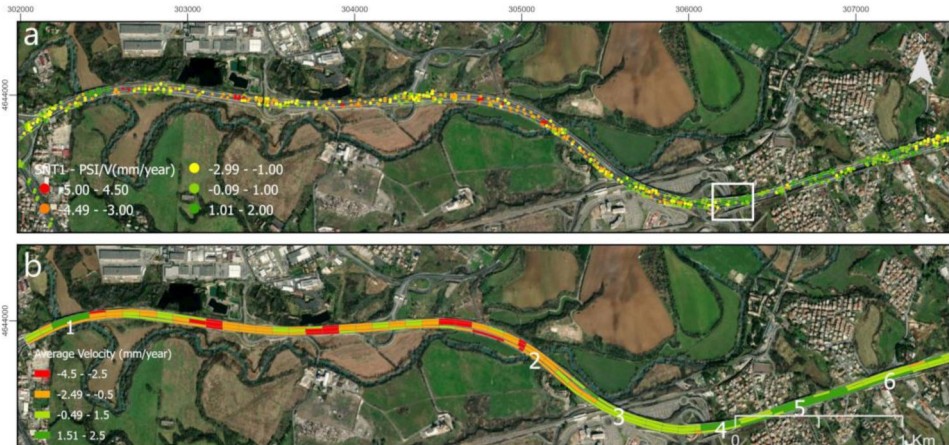

**Figure 10.** (**a**) Displacement map PSI Sentinel-1 A24 highway, (**b**) Map classification sections average velocity (mm/year) and bridge 1-6, as background ESRI source 2020.

In Figure 10a, a maximum displacement velocity of −4.8 mm/year was obtained, where this occurred in the embankments. In Figure 10b, the subsidence of the embankments is corroborated, while the series of highway bridge crossings along the analyzed section has good stability, see crossing 5 (Figure 11).

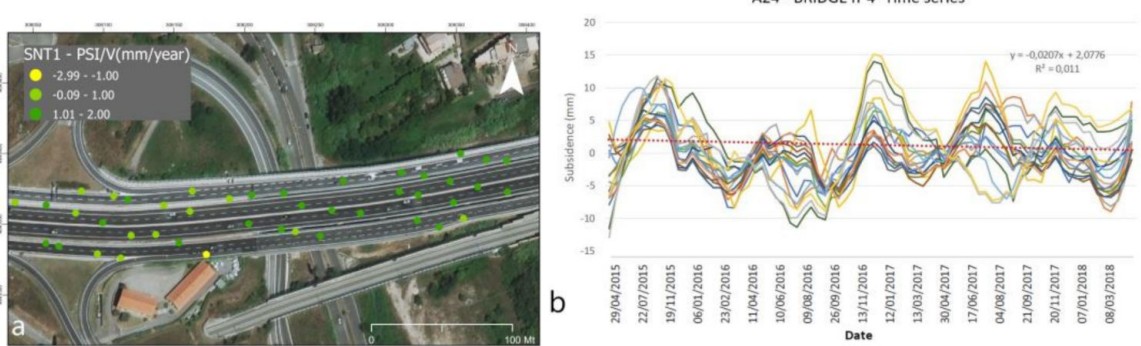

**Figure 11.** (**a**) Displacement map PSI Sentinel-1, A24 bridge-highway. As background orthophoto Rome, (**b**) Graph of the time series of bridge with structural bridge stability.

In the cases analyzed, the local geological conditions indicate that the loads of the highways may be one of the main factors of soil consolidation. However, further analysis is required to characterize subsidence-induced phenomena. By using the geological map of Lazio [57], we were able to calculate the average velocity rates for the each lithological type interested in by the highway network.

In Figure 12, gravel/sands/clay correspond to a low rate of subsidence for the PSI located along the highway network. As expected, unconsolidated deposits, e.g., sands and other alluvial materials, show higher subsidence rates compared to basement formations, such as marl and limestone.

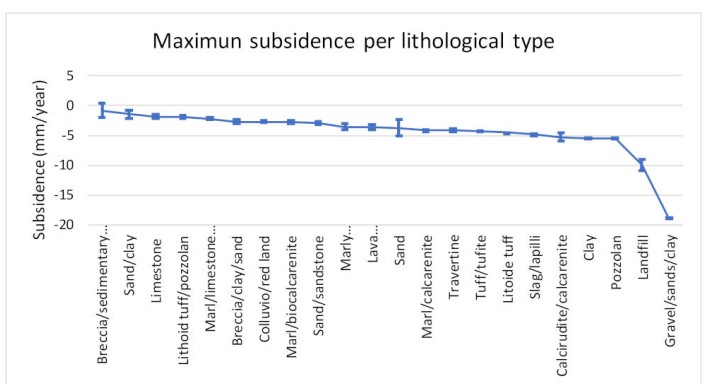

**Figure 12.** Graph subsidence per lithological type for highway network.

## 4. Results 2: Cosmo-SkyMed Data

### *4.1. High Resolution Analysis: A91 Highway*

With Cosmo-SkyMed, we deepened the analysis in the areas where we found subsidence with Sentinel-1, as evidenced in Figure 3b, box c. A detail of the subsidence and a study of the behavior of the embankment, located in an alluvial deposit, were obtained where there is a great displacement of the soil due to the underground water courses of the Tiber valley river towards the sea (see Figure 13b).

A91 Highway (Rome-Fiumicino)

For the A91 highway "Roma-Fiumicino", we used the high-resolution SAR data from the Cosmo-SkyMed observation period (see Table 1) to obtain an overview of the displacement velocities of the study area. We obtained a Displacement map PSI of the environment (see Figure 13a), then we proceeded to extract the PSI contained in the road route and classify them according to the average velocity entity as in the previous cases in sections of 100 mt listed according to the direction of travel (see Figure 14). With Cosmo-SkyMed, we reached ~18,000 PSI, where we can mainly show negative PSI values, with the velocity of some targets reaching more than −20 mm/year.

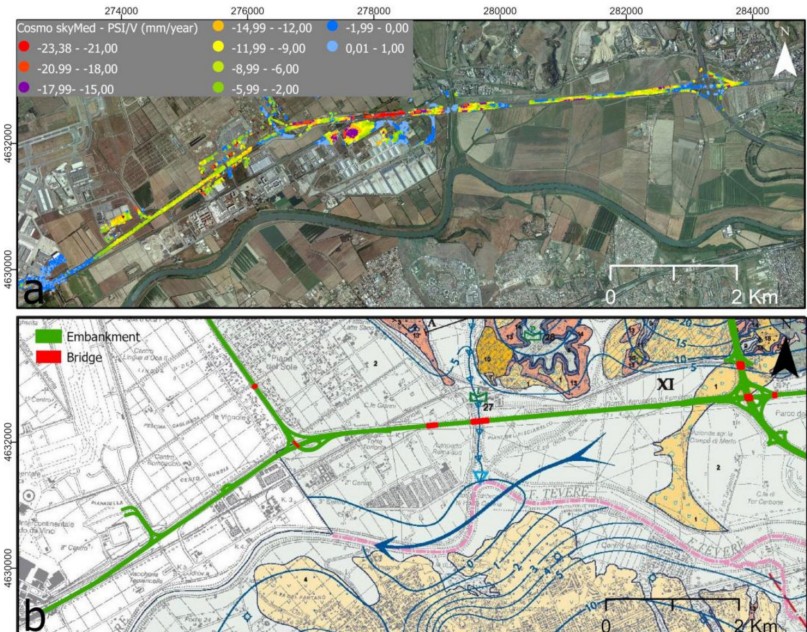

**Figure 13.** (**a**) Displacement map PSI Cosmo-SkyMed for Rome-Fiumicino, as background orthophoto Rome 2011, (**b**) Detail of the position of the A91 highway on the hydrogeological map of Rome (La Vigna et al., 2015) shows the position of the route in the alluvial deposit, indicates the flow of groundwater (blue arrow), indicates anthropogenic material due to hydraulic recovery works (mixed coffee).

To obtain the points contained in the carriageways and on the access roads of the A91, a map of the entire highway was drawn up (see Figure 14). We were able to calculate the average velocity (mm/year) of all the sections, and to later classify them into different classes according to the average velocity; here, most of the sections detected negative average velocity values, reaching a considerable amount of PSI for each section (see Figure 15a).

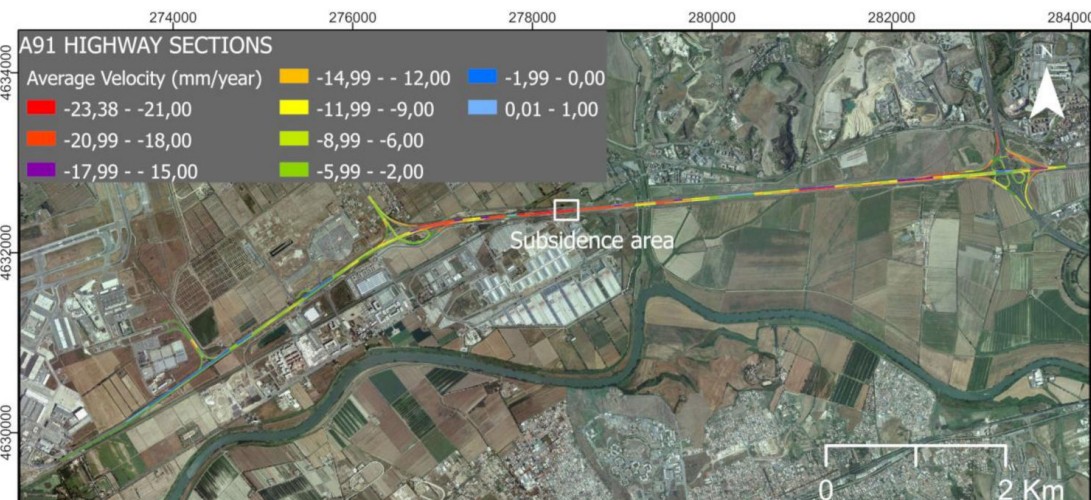

**Figure 14.** Map of sections of the A91 highway, classified by average velocity entity, as background orthophoto Rome.

The subsidence area for the A91 highway is shown in the white box in Figure 14. We enumerated the sections in the Roma and Fiumicino directions, and we detected the sections with the greatest displacement:

- S-39 and S-38, for the carriageway in the direction of Fiumicino; and
- S-60 and S-61 for the carriageway to Rome.

The largest displacement velocity of the S-39 section is −23 mm/year at 57 PSI, and the values of R2 are close to 1, showing a good fit between the estimated trend and the PSI data along the sections (see Table 3).

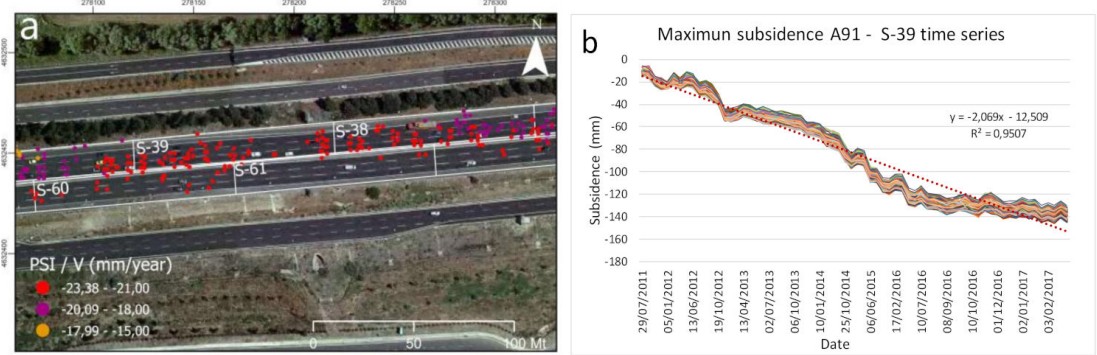

**Figure 15.** (**a**) Detail of the subsidence area of the A91 highway as background orthophoto Rome 2011. (**b**) Graph of the time series of section S-39.

The subsidence detected in the road network is mainly observed in embankment structures, both in the case of Sentinel-1 and in the case of Cosmo-SkyMed.

In the case of the subsidence sections of A91 (Figure 15), the major subsidence is confirmed for a certain lithological type: gravel/sans/clays, this can also be seen in the results 1 of Sentinel-1 (Figure 12). However, more research is required to characterize subsidence-induced phenomena.

**Table 3.** Sections road subsidence of the A91 highway.

| Sections | N° PSI | Average Velocity (mm/year) | Standard Deviation | R2 |
| --- | --- | --- | --- | --- |
| S-39 | 56 | −23.3 | 0.54 | 0.95 |
| S-38 | 71 | −21.2 | 0.91 | 0.89 |
| S-60 | 22 | −23.0 | 1.44 | 0.92 |
| S-61 | 5 | −23.1 | 1.03 | 0.92 |

In the 6-year observation, a target with high spatial density is detected, and in the sections of Table 3, they exceed −20 mm/year, with standard deviation between 0.5 and 1.5, and the R2 of each sections is close to 1, which implies that the adjustment between time and buckling measurements allows explanation of the fluctuation of the measurements.

From a theoretical point of view, the linear trend of subsidence is associated with the consolidation process over time. Therefore, a decrease in the displacement rate is expected over time during the consolidation process. In order to deepen this analysis, it would be necessary to study the consolidation process for this type of embankment structure built in a given time, studying the compressible units.

## 5. Conclusions

The DInSAR satellite interferometry techniques are useful to manage road operability on a large scale. Two interferometry techniques were used: the first based on a completely open-source SNAP-StaMPS workflow, with open SAR data on the Copernicus Open Access Hub the European Space Agency (ESA), and the second based on a Small Baseline Subset (SBAS) technique using a Cosmo-SkyMed dataset obtained by a License Agreement with the Italian Space Agency (ASI). The amount of data available from Sentinel-1 allowed investigation of the overall structural behavior of a transport network, in this case the highways network of Metropolitan Rome. With Cosmo-SkyMed, we were able to confirm the displacement trends along the alluvial deposits of the Tiber Valley, as identified by the Sentinel-1 dataset, providing at the same time more spatial details.

The results showed that highway deformation occurs primarily over correlated alluvial deposits and embankment structures. The amount of PS targets detected by Sentinel-1 is sufficient to provide valuable information on the general state of operation of the structures and could be useful for planning maintenance and simplifying inspection operations. In addition, the structural stability of reinforced concrete road bridges is confirmed.

This method is a useful tool for the road managers. Indeed, it offers a quick and economic way to identify the sections that need maintenance and define the intervention priority. The presented method can represent the first step in the road network management system. Indeed, it could be used to have an overview of the road network condition on a large scale and to plan in-depth surveys on the single sections.

**Author Contributions:** Conceptualization, M.A.M, P.D.M.; methodology, M.M., P.J.V.D.; software, J.M.D.B., M.F.; validation, P.J.V.D.; writing F.O.; supervision, M.A.M. All authors have read and agreed to the published version of the manuscript.

**Funding:** This research received no external funding.

**Acknowledgments:** We acknowledge Maria Rosaria Manzo e Manuela Bonano of IREA-CNR for CSK data SBAS processing performed thanks to the I.MODI Project funded by Horizon 2020—SME INSTRUMENT PHASE 2, Grant agreement No 720121. For Cosmo-SkyMed data: "Project carried out using COSMO-SkyMed ® Product © ASI (Italian Space Agency) provided under license of ASI.

**Conflicts of Interest:** ## References

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
