# Peer review of "DInSAR for Road Infrastructure Monitoring: Case Study Highway Network of Rome Metropolitan (Italy)"

_remotesensing, doi:10.3390/rs12223697_

Round 1
Reviewer 1 Report
The submitted paper presents a complex study combining displacements with rainfall and soil-type data; the selected journal for publishing the paper (Remote Sensing) is appropriate; the methods introduced here are novel and the paper deserves publishing. The paper is well written. The readers can benefit from the practical examples of the use of open-source software tools; the methods are listed and explained.
The introduction could be more concise, more easy-to-read, yet it is not necessary to make changes.
Particular suggestions for improvement are listed below:
Line 69: I believe that "namely" would be more appropriate than "nominally" here,
Line 118: remove the redundant period in the end,
Figure 8: please, provide the plot in better quality: separator of years/months should be a "/" not a ",", "TIME" should be changed to "Date" (not in all caps), the image resolution is poor (it would be appropriate to use vector graphics such as eps/pdf, etc.); the same remarks apply to Figures 11 and 14,
Lines 345-375: replace the instructions to authors with an actual listing of author contributions, funding sources, and conflicts of interest.
Conclusions: please, briefly describe the used methods (principles, accessibility, algorithms), results, and implications.
Concluding remark: comparing the measured data with other measurement methods would make the contribution much more valuable.
Author Response
Reviewer #1
The introduction could be more concise, more easy-to-read, yet it is not necessary to make
changes.
Particular suggestions for improvement are listed below:
Line 69: I believe that "namely" would be more appropriate than "nominally" here,
the word "nominally" is changed to "namely"
Line 118: remove the redundant period in the end,
Deleted Line [130-134].
Figure 8: please, provide the plot in better quality: separator of years/months should be
a "/" not a ",", "TIME" should be changed to "Date" (not in all caps), the image resolution
is poor (it would be appropriate to use vector graphics such as eps/pdf, etc.); the same
remarks apply to Figures 11 and 14,
The quality of the plots is improved and corrected for the Date references [1-15].
Lines 345-375: replace the instructions to authors with an actual listing of author
contributions, funding sources, and conflicts of interest.
The author contribution section was corrected
Conclusions: please, briefly describe the used methods (principles, accessibility,
algorithms), results, and implications.
A new paragraph to the conclusions was added [336-340].
Concluding remark: comparing the measured data with other measurement methods
would make the contribution much more valuable.
A reliability analysis using the PSI data and the geological map of Rome, to show the
maximum subsidence by lithological typology along the entire highway network. Added
line 319-323 and Figure 12.
Reviewer 2 Report
Proposed study seems to be very interesting and valuable, however it was difficult to judge whether your paper can be accepted or not, because many figures were difficult to interpret due to the lack of resolution, legend and small words which are difficult to read. I hope you revise these figures dramatically and submit again. Please see the details of questions and comments as follow.
[1] P2L64,“Although〜network [27,28]”: Since there are few cases of observing time-series changes in large-scale structures in previous studies, I think your analysis is very valuable. However, according to the explanation in the “introduction” part, it seems that similar studies have been carried out in the past (ex. Reference [27] and [28]). Comparing with these similar studies, could you clarify the novelty of your study?
[2] Table 1:Please specify not only the information of first observation and final observation, but also show the information of other observations.
[3] All figures from figure 1 - 14:It is difficult to interpret the figure as a whole due to the low resolution, lack of legends or they were not described properly. Please improve them as follow;(1) If the resolution is low, please improve to high resolution.(2) If there is no legend in the figure, please add it.(3) Some words in legends were difficult to read, because white color was assimilated with the background images (ex. Figure 5). Please improve them. (4) If you want to compare the left and right figures as shown in Fig. 7, please use the same range of values ​​for each rank in legend.
[4] Table 3: I think it is a very interesting result to find the possibility of understanding the subsidence of the road from long-term observation data. However, in order to properly show the reliability of the observation data, it needs a verification using on-site measurement data. If you have those data, compare them with your analyzed result and verify it. If you do not have on-site data, please discuss and add some explanations about reliability of your result in the discussion section.
[5] Figure 14: Final table seems to show interesting result. However, the relationship of R2 value of the linear regression and the reliability of subsidence are not clear. Please add some explanations and show the reliability of subsidence observation.
Author Response
Reviewer #2
[1] P2L64,“Although〜network [27,28]”: Since there are few cases of observing time-series changes in large-scale structures in previous studies, I think your analysis is very valuable.
However, according to the explanation in the “introduction” part, it seems that similar studies have been carried out in the past (ex. Reference [27] and [28]). Comparing with these similar studies, could you clarify the novelty of your study?
Our work is one of the first extensive application of Sentinel 1 data focusing on linear infrastructures covering a very large area. Previous studies are focused the setting up of web platforms for managing transportation systems including also INSAR derived information [27] or only on the exploitation of COSMO-SkyMed data [28].
[2] Table 1: Please specify not only the information of first observation and final
observation, but also show the information of other observations.
The information is valid for the whole datasets
[3] All figures from figure 1 - 14:It is difficult to interpret the figure as a whole due to the low resolution, lack of legends or they were not described properly. Please improve them as follow;(1) If the resolution is low, please improve to high resolution.(2) If there is no legend in the figure, please add it.(3) Some words in legends were difficult to read, because white color was assimilated with the background images (ex. Figure 5). Please improve them. (4) If you want to compare the left and right figures as shown in Fig. 7, please use
the same range of values for each rank in legend.
All the figures were checked, the resolution of the figures of [1-14] was increased, the quality of the graphics was improved, changing TIME per Date, and giving uniformity to each one of them.
A dark background was added to each of the figures that presented little visualization of the legends, figures [5,7,11,13 and 14].
In the case of figures 5 and 7 the classification was maintained separated since it is referred to a further aggregated classification of the filtered PS dataset.
[4] Table 3: I think it is a very interesting result to find the possibility of understanding
the subsidence of the road from long-term observation data. However, in order to properly show the reliability of the observation data, it needs a verification using on-site measurement data. If you have those data, compare them with your analyzed result and verify it. If you do not have on-site data, please discuss and add some explanations about reliability of your result in the discussion section.
Our work is aimed at highlighting the capability of DInSAR dataset of collecting usefulinformation on long-term deformation processes that can be attributed soil consolidation
processes in case of road built up over embankments loading soft soils, and, conversely, at identifying the presence of instability that cannot be explained by the local geological context. In this respect, the analysis has the pure objective of identifying undetected criticisms at very large scale and, at this stage, rely only on the correlation between the observed displacements and the characteristic of the foundation soils.
An attempt to quantitatively prove the reliability of the first Sentinel 1 results is made by
inserting a new figure graph indicating the displacement (subsidence) in function of the
lithological types. This graph was made by crossing the PSI data of the motorway
network with the geological map of the Rome region Lazio, scale 1: 25,000. Added line
326-334.
A further check can be obtained by comparing the results from Sentinel 1 data with those
from CSK data that furnished similar rate of displacement along the same road segment
(see S-39 in figure 15 e Table 3)
5] Figure 14: Final table seems to show interesting result. However, the relationship of
R2 value of the linear regression and the reliability of subsidence are not clear. Please add
some explanations and show the reliability of subsidence observation.
In the study it is shown that subsidence is mainly evidenced in embankment structures,
close to water courses. Currently we do not have empirical evidence of this phenomenon,
but we clearly showed that it is correlated with the lithological type. This is confirmed in
by both datasets, Sentinel 1 and CSK. Added line 319-323 and figure 12.
Round 2
Reviewer 2 Report
Thank you for modifying the manuscript according to the review comments.
Quality of presentation has been improved and some discussions were added regarding of the interpretation of the results. Therefore, the paper can be accepted.